# Neuronal loss of the pentose phosphate pathway in the living nervous system is causally linked to [NADPH] reduction and elevated oxidative stress

Stephan Müller[1], Nina Surina[1], Andrés Köhler-Solís[1], Ioannis Nellas[1], Astrid Fleige[2], Sebastian Görtz[2] and Stefanie Schirmeier[1]

[1]*Zoology and Animal Physiology, Faculty of Biology, Technische Universität Dresden, Dresden, Germany*
[2]*Institute for Neuro- and Behavioral Biology; Faculty of Biology, Universität Münster, Münster, Germany*

Handling Editors: Katalin Toth & Valentina Mosienko

The peer review history is available in the Supporting Information section of this article (https://doi.org/10.1113/JP288582#support-information-section).

The Journal of **Physiology**

**Abstract figure legend** Using genetically encoded sensors for NADPH and $H_2O_2$ in the living *Drosophila* nervous system, neuron-specific PPP knockdown is shown to result in reduced neuronal NADPH levels and elevated neuronal $H_2O_2$ levels and oxidative stress. Further neuronal PPP knockdown induces progressive neurodegeneration and neuronal dysfunction.

**Abstract**   Neurons are highly specialized cells that require large amounts of energy to function. Glial cells support neurons in many ways, including metabolically. In *Drosophila*, neuronal glycolysis has been found to be dispensable, as long as glial glycolysis is intact, a finding supporting a conservation

of the astrocyte-neuron-lactate shuttle (i.e. ANLS)-hypothesis. Neurons use glia-derived lactate to fuel their highly oxidative metabolism. Nevertheless, they readily take up glucose. It has been hypothesized that neuronal glucose might be pre-dominantly metabolized through the pentose phosphate pathway (PPP) rather than glycolysis to produce reduction equivalents in the form of NADPH to cope with the oxidative stress caused by a highly oxidative metabolism and prevent oxidative damage. We show that knockdown of components of the PPP in all neurons in *Drosophila* induces mild neurodegeneration, which can be rescued by antioxidant feeding. To directly link a putative loss of neuronal NADPH to elevated reactive oxygen species (ROS), we generated fly lines expressing biosensors for NADPH and $H_2O_2$ and developed methods to image the sensors in *Drosophila* neurons. Panneuronal PPP knockdown results in reduced neuronal NADPH and elevated $H_2O_2$ levels in larval tissue. In addition, multiparametric live imaging of fully differentiated neurons in the adult *Drosophila* brain shows decreased NADPH levels and increased ROS stress upon PPP knockdown. Even though the phenotypic consequences of elevated ROS are mild, these data demonstrate that loss of PPP, reduced NADPH levels and increased oxidative stress are indeed functionally linked in living tissue.

(Received 20 January 2025; accepted after revision 18 February 2026; first published online 15 March 2026)

**Corresponding author** S. Schirmeier: Zoology and Animal Physiology, Faculty of Biology, Technische Universität Dresden, 01217 Dresden, Germany.     Email: stefanie.schirmeier@tu-dresden.de

### Key points

- Neuronal pentose phosphate pathway (PPP) knockdown induces neurodegeneration that can be rescued by food-derived antioxidants.
- Neuronal PPP deficiency results in reduced neuronal NADPH levels in living tissue.
- Neuronal PPP deficiency results in elevated neuronal $H_2O_2$ levels in living tissue and oxidative stress.

## Introduction

Neurons require a large amount of energy to sustain physiological function. Therefore, they are metabolically supported by glial cells, which are highly glycolytic cells that produce lactate that is shuttled to the neurons (Fünfschilling et al., 2012; Hall et al., 2012; Lee et al., 2012; Mächler et al., 2016; Volkenhoff et al., 2015; Bonvento & Bolaños, 2021; Magistretti & Allaman, 2022). In the neurons, lactate is used to fuel a highly oxidative energy metabolism (Hall et al., 2012; Bonvento & Bolaños, 2021; Magistretti & Allaman, 2022). This metabolic coupling between the glial cells and neurons is evolutionarily conserved and can be found from flies to mammals

(Pellerin & Magistretti, 2012; Volkenhoff et al., 2015). In *Drosophila*, neuronal glycolysis is even dispensable; nonetheless, neurons take up glucose (Volkenhoff et al., 2015, 2018). Because a highly oxidative metabolism leads to the generation of reactive oxygen species (ROS), it has been hypothesized that glucose metabolism via the oxidative reactions of the pentose phosphate pathway (PPP) rather than glycolysis might be essential in neurons and used to produce reduction equivalents in the form of NADPH to cope with oxidative stress and prevent oxidative damage (Herrero-Mendez et al., 2009 reviewed in Bonvento & Bolaños, 2021; Magistretti & Allaman, 2022). It has been difficult to assess how high the glucose flux through the PPP really is in neurons because glucose 6-phosphate can

**Stephan Müller** completed his Bachelor studies in Biology at Friedrich Schiller University Jena, Germany. After that, he obtained a Master's degree in Integrative Neuroscience at University of Magdeburg (Germany). After his studies, he joined the group of Stefanie Schirmeier, first at University of Münster, later at Dresden University of Technology, to conduct his PhD work entitled 'Alternative Glucose Metabolization Prevents ROS-Induced Progressive Neurodegeneration'. He is currently working as a field service engineer for Yokogawa.

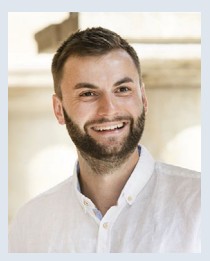

be recycled from the PPP and thus many studies probably underestimate flux through the PPP (Bouzier-Sore & Bolaños, 2015).

In cells, two major sources of ROS have been described, NADPH-oxidases and the electron transport chain, contributing to ~85% of all ROS (Parascandolo & Laukkanen, 2019; Sies & Jones, 2020). Additionally, there are at least 40 known enzymes in humans that can contribute to ROS formation, whose contributions seem to vary strongly depending on the compartment/organelle (Sies & Jones, 2020). Cytosolic ROS clearance mainly depends on the oxidation of glutathione (2GSH → GSSG). Two molecules of NADPH are needed to reduce GSSG back to GSH, a reaction that is essential for replenishing the GSH pool, especially in neurons, in which the GSH pool is limited (Bolaños et al., 1996; Dringen et al., 1999; reviewed in Aoyama, 2021). The NADPH needed for maintaining the GSH pool is considered to mainly be replenished via the oxidative phase of the PPP. The PPP is the second major glucose metabolizing pathway; it is divided into a non-reversible, oxidative, branch and a reversible, reductive, branch. It is tightly inter-connected with glycolysis, sharing a common pool of glucose 6-phosphate. The PPP has been described to play a role in a wide range of cell biological processes, from the prevention of oxidative damage, tumour growth and lipid synthesis, to sugar appetite, in addition to learning and memory, as well as circadian rhythms (Agrawal & Canvin, 1971; Carvalho-Santos et al., 2020; de Tredern et al., 2021; Herrero-Mendez et al., 2009; Jiang et al., 2011; Rey et al., 2016). Recently, the oxidative branch came into focus as a cellular redox-detection mechanism. Originally described in yeast, it is now widely accepted that glucose is shuttled into the PPP via ROS-induced inhibition of glycerol aldehyde 3-phosphate dehydrogenase, a key glycolytic enzyme (Krüger et al., 2011; Ralser et al., 2009). Indeed, studies ranging from mice to *Drosophila*, hypothesized a beneficial role of increased glucose 6-phosphate dehydrogenase activity as a coping mechanism for oxidative stress (Della Noce et al., 2019; Jeng et al., 2013; Kuehne et al., 2015; Legan et al., 2008; Nóbrega-Pereira et al., 2016).

In the present study, we investigate the importance of neuronal PPP to maintain NADPH levels and ROS homeostasis. We show that pan-neuronal knockdown of genes encoding enzymes involved in NADPH production via the PPP induces mild progressive neurodegeneration, comprising a phenotype that can be rescued via dietary antioxidants. We generated fly lines expressing biosensors for NADPH and $H_2O_2$ (Ermakova et al., 2014; Tao et al., 2017), allowing measurement of both metabolites in the living brain. Using these sensors, we show that loss of neuronal PPP indeed alters NADPH levels and induces $H_2O_2$ accumulation, demonstrating a functional link between the PPP and oxidative stress.

## Methods

### Fly work

Although no additional working permissions are required for this animal model, all experiments have been designed to avoid unnecessary hardships for the animals. All S1 experiments are approved by local authorities (54-8451/178).

Unless otherwise noted, fly stocks were kept on standard fly food under a 12:12 h light/dark photocycle at 25°C.

Crosses including *tub-Gal80ts* were raised on 18°C and put at permissive temperature of 29°C after 4 weeks. For antioxidant experiments, standard fly food was supplemented with 0.36 mM L-ascorbic acid and 0.46 mM α-tocopherol. Mated female flies were flipped on fresh medium including all supplements every 2–3 days.

Transgenic animals were generated as follows: the iNap1 containing plasmid was generously gifted by Tao et al. (2017). The HyPerRed containing plasmid was generously gifted by Ermakova et al. (2014). The coding regions were PCR amplified and cloned into pENTR/D-TOPO (using 5′ CACC forward primers; HyPerRed forward: caccATGGAGATGGCGAGCC AGCA, reverse: TCATTAAACCGCCTGTTTTAAAAC; iNap1 forward: caccTGACGTCAATGGGAGTTTGT, reverse: GATGGCTGGCAACTAGAAGG). Afterwards, the coding sequences were cloned via gateway cloning (Gateway LR; Invitrogen, Waltham, MA, USA) into the vector pUASTattBrfa (Bischof et al., 2013), which allows ΦC31 integrase-mediated integration into the fly genome. The resulting vectors have been integrated into the fly genome at landing site attP2 and attP40. All other transgenic animals used in this study can be found in Table 1.

### Immunohistochemistry

Immunohistochemistry was performed as described in (Andlauer et al., 2014). In brief, after fixation (4% paraformaldehyde in phosphate-buffered saline Triton X-100 (PBT), 25 min), adult female brains were incubated with primary antibodies (rat-α-*N*-cadherin 1:5; DSHB DN-Ex #8; Developmental Studies Hybridoma Bank, Iowa City, IA, USA) in 5% normal goat serum in phosphate-buffered saline (PBS) for 48 h at room temperature and then washed in PBT for 3 h, followed by overnight incubation with secondary antibodies (α-rat-Alexa568; Thermo Fisher Scientific, Waltham, MA, USA) in PBT at 4°C. The brains were then washed for 3 h with PBT and mounted in VectaShield (Vector Laboratories, Burlingame, CA, USA). To observe the levels of oxidized lipids, 20-day-old fixed female brains were incubated with the antibody against 4-hydroxynonenal

**Table 1. Transgenic fly lines used in the present study.**

| transgene | sourcs |
|---|---|
| G6Pdh-KK | v101507; dsRNA targeting G6Pdh |
| G6Pdh-TRIP | BL50667; dsRNA targeting G6Pdh |
| G6Pdh-GD | v3337; dsRNA targeting G6Pdh |
| Pgd-KK | v100269; dsRNA targeting Pgd |
| Pgd-TRIP | BL65078; dsRNA targeting Pgd |
| kl-3-GD | v32971; dsRNA targeting a male fertility factor, used as negative control |
| kl-5-TRIP | BL55609; dsRNA targeting a male fertility factor, used as negative control |
| GFP-dsRNA | BL9331; mock dsRNA, used as negative control |
| UAS-iNap1$^{attp40}$ | Present study; plasmid from Tao et al. (2017) |
| UAS-iNap1$^{attp2}$ | Present study; plasmid from Tao et al. (2017) |
| UAS-HyPerRed$^{attp40}$ | Present study; plasmid from Ermakova et al. (2014) |
| UAS-HyPerRed$^{attp2}$ | Present study; plasmid from Ermakova et al. (2014) |
| UAS-pHerry | Rossano et al. (2017) |
| UAS-GFP | BL1521 |
| UAS-mcherryCAAX$^{86Fb}$ | A. Volkenhoff, C. Klämbt |
| Elav-Gal4 (I) | Lin and Goodman (1994) |
| Elav-Gal4 (III) | Lin and Goodman (1994) |
| Repo-Gal4(II) | Lee and Jones (2005) |
| Repo-Gal4(III) | Sepp and Auld (1999) |
| Eyeless-Gal4 | Domínguez and de Celis (1998) |
| tubGal80ts | BL7018 and BL7019, McGuire et al. (2004) |

*Note*: Knockdown efficiency of *G6Pdh*- and *Pgd-dsRNA* lines was assessed via quantitative real-time PCR (Fig. 1).

($\alpha$ 4-HNE; #46545; rabbit; dilution 1:100; Abcam, Cambridge, UK) (as previously described in Bailey et al., 2015), together with $\alpha$ NCad (DSHB DN-Ex #8; rat; dilution 1:10; Developmental Studies Hybridoma Bank). Immunofluorescence staining was performed as described above.

### Semi-thin-sections

Heads of adult flies were embedded in Epon as described previously (Stork et al., 2008). Then, 1 μm semi-thin sections were cut using a EM UC7 microtome (Leica, Wetzlar, Germany), stained with toluidine blue and imaged using an Axiophot microscope (Zeiss, Oberkochen, Germany).

### *Drosophila* activity monitoring

Mated females were aged for 15 or 30 days on standard food at 29°C. Flies were loaded into individual capillaries containing standard food. Capillaries were placed in a *Drosophila* activity monitor (DAM5H; TriKinetics, Waltham, MA, USA) under a 12:12 h light/dark photocycle at 29°C. The animals were acclimated to the vials and incubator for 24 h and then the activity of each fly was recorded over a 48 h period. Activity was binned into 20 min intervals during measurements. For analysis, the mean activity per time interval of the 2 days was calculated. Activity was binned into 1 h intervals for visualization in Fig. 3*A*. The mean activity for different time intervals [daytime activity (ZT0-9), siesta (ZT9-12), evening activity peak (ZT12-16) and night rest (ZT16-24)] was calculated and statistically significant difference in total moves was determined using the non-parametric Kruskal–Wallis test with Dunn's multiple comparison.

### Imaging and data analysis

**Imaging of fixed samples.** All images of fixed tissue were obtained using a Leica SP8 DMi8 inverted microscope or an AxioObserver LSM880 inverted microscope (Zeiss). Imaging for holes quantification (Fig. 2*C* and *D*) was performed using a HC L APO CS2 40×/1.3 oil objective (Leica) at a resolution of 1024 × 1024 pixels with a 1 μm z-step size. *N*-cadherin was labelled with Alexa568, excited at 552 nm and recorded at 560–620 nm. The beam splitter used was DD 488/552. The pinhole was set to 1 AU. Fixed samples were imaged at an AxioObserver LSM880 inverted microscope (Zeiss, Germany) using immersion objectives (Figure 2*C*). Samples were imaged with a 40× objective at a resolution of 1024 × 1024 pixels with a z-step size of 1 μm. NCad (labelled with Alexa594) was excited with the 594 nm laser and acquisition was performed at 590–600 nm. The beam splitter used was DD 405/594. The pinhole was set to 1 AU. Adult neurodegeneration experiments were analysed manually. All samples were

randomized and re-named numerically. Holes were then counted manually, blindly and samples were decrypted afterwards. Analysis was performed using Prism, version 10.3.1 (GraphPad Software Inc., San Diego, CA, USA).

To oxidized lipids samples were imaged with a 63× objective (2.0 digital zoom) at a resolution of 1024 × 1024 pixels with a z-step size of 0.2 μm up to a total of 10 μm. 4-HNE (labelled using Alexa488) was excited at 488/acquisition 488–500 nm, NCad (labelled with Alexa 594) was excited at 594 nm/acquisition 590–620 nm, whereas 4′,6-diamidino-2-phenylindole (DAPI) was excited with diode 405/acquisition 405–500 nm. The beamsplitter used was DD 405/488/594, with pinhole set at 1 AU. Images of mushroom body calyxes were segmented, using the DAPI label to define the ROI of the calyx, and the mean grey value of NCad and 4-HNE staining was quantified in this segmented region, and then tabulated in Excel (Microsoft Corp., Redmond, WA, USA). Fluorescence values are presented as a ratio of the raw signal of 4-HNE over NCad in each calyx obtained per brain. Analysis was performed using Prism, version 10.3.1 (GraphPad Software Inc.). Three biological replicates were analysed for each experimental condition, with four brains with their two respective calyxes each. The results presented correspond to 4-HNE/NCad ratio per calyx. After reviewing outliers and normality tests, statistical comparisons were performed as comparisons *vs.* control (*GFP^{dsRNA}*) with the Kruskal–Wallis test and with Dunn's *post hoc* test.

**Live imaging.** Larval eye imaginal disc live imaging was performed on a SP8 DMi8 inverted microscope (Leica), using a HC PL APO CS2 20×/0.75 dry or a ACS APO 20×/0.6 multi-immersion objective. Larval brains with the imaginal discs attached were dissected in fresh, pre-warmed HL-3 buffer without calcium and trehalose (70 mM NaCl, 5 mM KCl, 20 mM MgCl$_2$•6H$_2$O, 10 mM NaHCO$_3$, 115 mM sucrose and 5 mM Hepes) and mounted on poly-L-lysin coated cover slips and placed in a custom-made flow-through live imaging chamber. For HyPerRed live imaging, after 10 min, HL-3 was replaced by HL-6 (Macleod et al., 2002) buffer containing 600 μM H$_2$O$_2$ (CP26.1; Carl Roth, mannheim, Germany) and 10 mM digitonin (D141; Sigma, St Louis, MO, USA) for 5 min, followed by a 10 min incubation with HL-6 containing 10 mM pyruvate (P5280; Sigma) and 20 mM digitonin. The frame rate was 10 s. HyPerRed was excited at 552 nm and recorded at 560–620 nm; GFP was excited at 488 nm and recorded at 500–530 nm. The beam splitter used was DD 488/552. The resolution was 512 × 512 pixels for 8-bit images. The pinhole was set to 5 AU. For iNap1 live imaging, after 10 min, HL-3 was replaced by HL-6 buffer containing 100 μM cumene hydroperoxide (CHP; 247502; Sigma) and 10 mM digitonin for 2 min, followed

by a 10 min incubation with HL-6 containing 40 μM NADPH (481973; Millipore, Burlington, MA, USA) and 70 mM digitonin. Framerate was 10 s. iNap was excited at 405 nm and recorded at 505–525 nm; mCherry^{CAAX} was excited at 552 nm and recorded at 560–620 nm. The beam splitter used was DD 488/552. Laser intensities were adapted according to the expression levels, but kept constant compared to controls. The resolution was 256 × 256 pixels for 8-bit images. The pinhole was set to 5 AU. For pH imaging, pHerry expressing eye imaginal disc neurons were imaged for 20 min at a 10 s interval at three different pH-conditions (7.0, 7.5 and 6.5) as previously described (Rossano et al., 2017). pHerry live imaging mCherry was excited at 552 nm and recorded at 560–620 nm, pHlourin was excited at 488 nm and recorded at 500–530 nm. The beam splitter used was DD 488/552. The resolution was 512 × 512 pixels for 8-bit images. The pinhole was set to 5 AU.

Adult live imaging experiments were performed on an upright Axio.ImagerZ2 epi-fluorescence microscope (Zeiss) using a W Plan-Apochromat 40x/1.0 DIC M27 objective (Zeiss) and water immersion. Samples were excited using a xcite xylis white light LED (Excelitas Technologies Corp., Waltham, MA, USA). LED intensity was set to 25% at 10 ms (iNap1) and 40 ms (HyPer-Red) excitation time. Excitation filters were at 405 nm and 500 nm, for iNap1 and HyPerRed, respectively. Emission wavelengths were recorded using a specially designed filter cube (AHF Analysetechnik AG, Tübingen, Germany) allowing for dual band pass transmission. Images were recorded using an Axiocam 705 mono (Zeiss) in 14-bit mode at full resolution. ROIs were drawn manually around antennal lobes and raw intensity was measured without any form of correction.

### RNA extraction and qPCR

Knockdown efficiency of *G6Pdh-* and *Pgd-dsRNA* lines has been assessed via quantitative real-time PCR (qRT-PCR) (Fig. 1). RNA from 10 adult brains of 5-day-old female flies was isolated and purified following (Rio et al., 2010). DNA digestion, cDNA synthesis and qRT-PCR were performed following standard protocols. The oligonucleotides used as primers are listed in Table 2. Results were analysed using $\Delta\Delta$Ct method.

### Statistical analysis

The tests used for statistical analysis of the different data sets are reported in the Methods section and where appropriate. A file summarizing all information on the statistical analyses is available online (https://github.com/StefanieSchirmeier/statistics-data-Muller-et-al.2025).

**Table 2. Oligonucleotides.**

| primer | sequence |
|---|---|
| G6Pdh_fw | CGGCAAGATTCCGCACACGTT |
| G6Pdh_rev | TGCTCGTGCGGCTGGACCTT |
| Pgd_fw | TCGTGGTGTGCGCCTACAACC |
| Pgd_rev | AGTCGTCGACTGCACTTCCAG |

## Results

### Neuronal PPP knockdown induces progressive neurodegeneration that can be rescued via dietary antioxidants

Elevated oxidative stress induces progressive neurodegeneration. To test whether the PPP is essential in fully differentiated neurons in the adult animal to cope with oxidative stress, we assessed the rate of neurodegeneration. Accordingly, we knocked down the genes encoding the two enzymes of the PPP involved in NADPH production, glucose 6-phosphate dehydrogenase (G6Pdh) and 6-phosphogluconate dehydrogenase (Pgd), specifically in neurons of the adult animal using RNA interference. To avoid developmental defects, we employed the TARGET system to express the dsRNA constructs only after eclosion (McGuire et al., 2003). To assess neurodegeneration, we analysed brain morphology using semi-thin-sections of heads of 15- and 30-day-old animals. Because the optic lobes are a highly structured region of the adult brain that allows easy assessment of neurodegeneration, we analysed these regions specifically (Fig. 2). The optic lobes of animals with a neuronal PPP knockdown (elav-Gal4 driven G6Pdh^dsRNA or Pgd^dsRNA) exhibit more holes, a hallmark of neurodegeneration, than control animals (Fig. 2*A*). This phenotype becomes more pronounced in aged animals, indicating a progressive effect of PPP loss. To verify these data, we used an additional approach that allows analysis of the whole volume of the optic lobes. We stained brains for *N*-cadherin, marking all neurons, and analysed the whole volume of the optic lobes via confocal microscopy (Fig. 2*B*). Also here, we observe a progressive increase of holes and thus neurodegeneration (Fig. 2*B*). Because we hypothesized that accelerated neurodegeneration in neuronal PPP knockdown animals is caused by elevated ROS levels, we assessed a possible neuro-protective function of antioxidants (Fig. 2*C* and *D*). We fed the flies on standard fly food supplemented with 0.36 mm

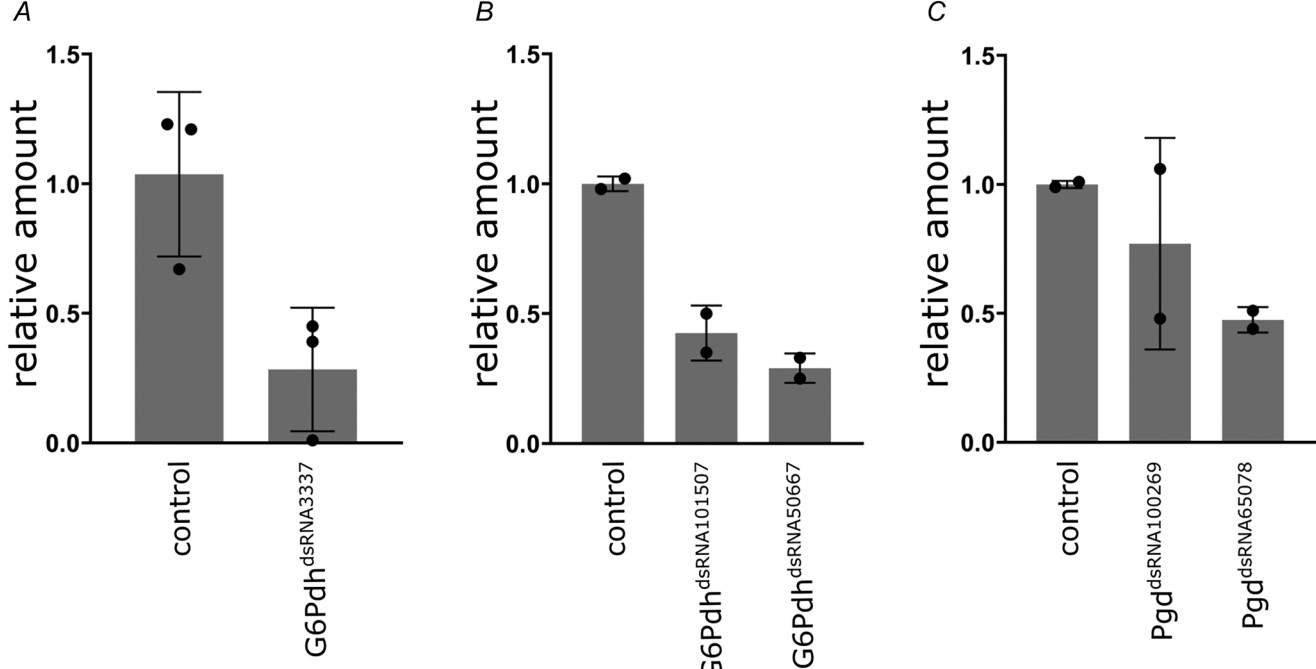

**Figure 1. qRT-PCR reveals efficiency of dsRNA-mediated knockdowns**
dsRNA-constructs were expressed in adult neurons using the following driver combination: *elav-Gal4^attP40*; *elav-Gal4^attP2*, *tub-Gal80^ts*. Knockdown efficiency of *G6Pdh^dsRNA3337* (A), *G6Pdh^dsRNA101507* and *G6Pdh^dsRNA50667* (B), *Pgd^dsRNA100269* and *Pgd^dsRNA65078* (C) was measured in whole brain lysates. Compared with the negative control (*kl-3^dsRNA32971*), the reduction of mRNA levels varies between 35% and 78% for the different dsRNA-constructs. This indicates a robust reduction of neuronal transcript expression, especially since glial expression is wild-typic and thus probably masking the full extent of neuronal knockdown. *N* = 2–3, *n* = 9 brains per sample.

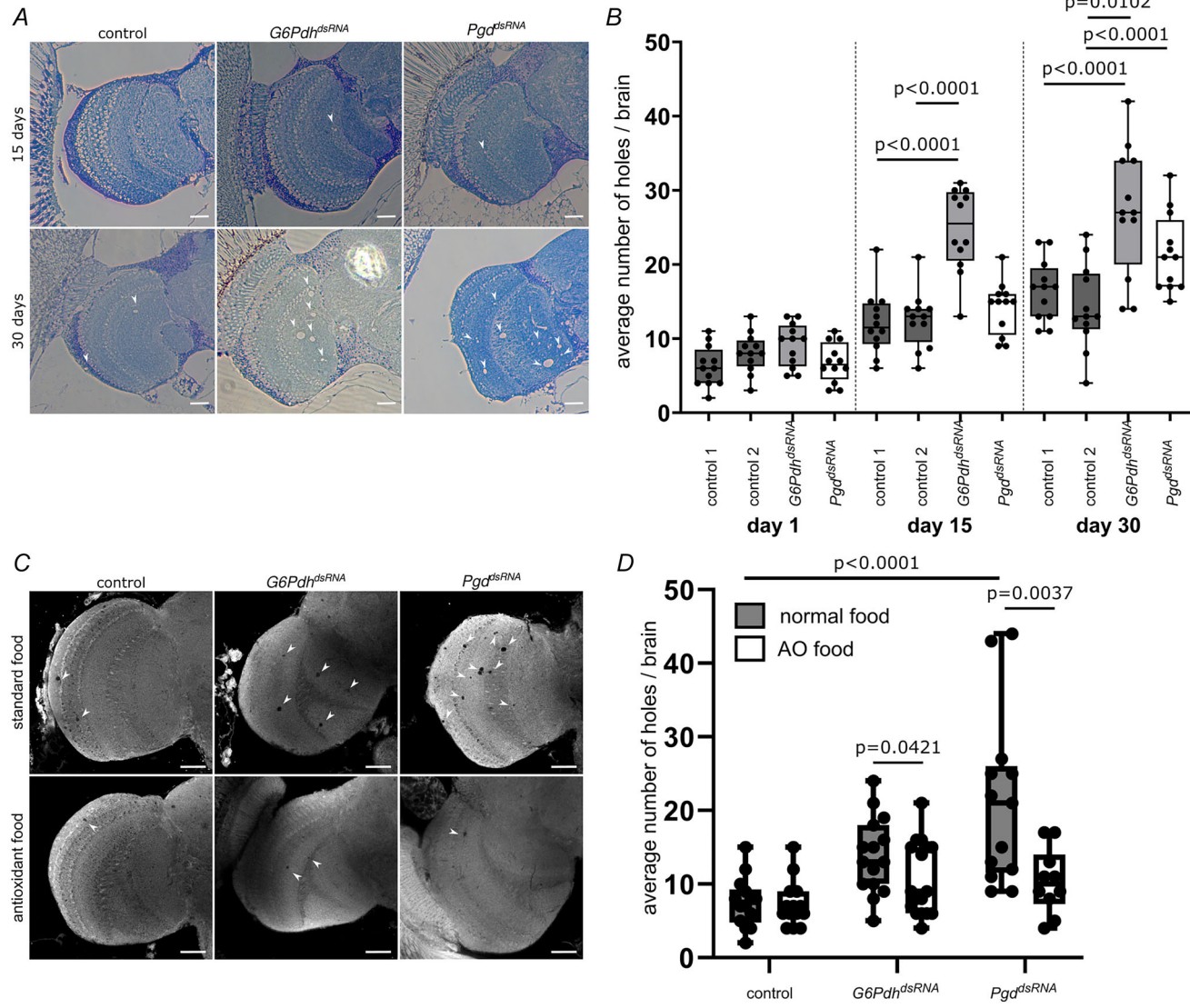

**Figure 2. Adult brains display progressive neurodegeneration upon neuronal PPP knockdown, which can be rescued by antioxidant treatment**

*A*, 15 days after hatching (upper row), the optic lobes of all genotypes show no signs of neurodegeneration. At the age of 30 days, in contrast, an accumulation of holes in the optic lobes of animals neuronally expressing *G6Pdh*[dsRNA101507] or *Pgd*[dsRNA100269] can be seen compared to control animals (expressing *kl-5*[dsRNA55609]). Arrows indicate holes. Representative images are shown. Scale bar = 40 μm. $N \geq 3$, $n \geq 3$. *B*, quantification of progressive neurodegeneration in adult brains using $\alpha$ NCad staining (N-Cadherin; stains all neurons). A progressive increase in the number of holes can be seen over time in *G6Pdh* and *Pgd* knockdown animals. $N = 3$, $n = 12$. dsRNA lines used: *GFP*[dsRNA9331] (control 1; mock dsRNA), *kl-3*[dsRNA32971] (control/ control 2), *G6Pdh*[dsRNA50667], *Pgd*[dsRNA65078]. Significance level between genotypes at the same age was determined using ANOVA with Tukey's multiple comparisons. *C*, confocal images of optic lobes stained with $\alpha$ NCad of 30-day-old animals that have been fed standard food (upper row) or antioxidant-enriched food (0.36 mm vitamin C and 0.46 mm vitamin E; lower row). Feeding on standard fly food supplied with antioxidants rescues neuronal PPP knockdown induced neuro-degeneration, white arrows indicate position of holes. Scale bar = 40 μm. $N \geq 2$, $n \geq 12$. dsRNA lines used: *kl-3*[dsRNA32971] (control), *G6Pdh*[dsRNA50667], *Pgd*[dsRNA65078]. *D*, quantification of holes indicating neurodegeneration. Neuronal knockdown of PPP induces neurodegeneration. This phenotype can be rescued by feeding animals on antioxidant-enriched food. Boxes display median, first and third quartile, whiskers extend to most extreme data points (1.5 × interquartile range) and outliers (circles). significance level between feeding conditions of one genotype was determined using the Welch two sample *t* test; significance level between genotypes at the same feeding condition was determined using ANOVA with Tukey's multiple comparisons. *P* values for significantly different conditions are shown; *P* values for non-significantly different conditions are not shown. $N \geq 2$, $n \geq 12$. dsRNA lines used: *kl-3*[dsRNA32971] (control), *G6Pdh*[dsRNA50667], *Pgd*[dsRNA65078].

ascorbic acid (vitamin C) and 0.46 mм alpha-tocopherol (vitamin E). Indeed, feeding antioxidant food over a time period of 30 days significantly ameliorates the neurodegenerative effects of neuronal knockdown of *G6Pdh* and *Pgd* (Fig. 2*C* and *D*). These data indicate that neuronal PPP loss induces oxidative stress in neurons, which leads to progressive neurodegeneration.

## Neuronal loss of PPP induces progressive reduction of maximal activity and circadian rhythmicity

It has been shown previously that neurodegenerative phenotypes characterized by neuronal loss, similar to the one described here, induce a progressive loss of climbing ability in *Drosophila* (Cabirol-Pol et al., 2017; Hindle et al., 2017; Sunderhaus et al., 2019). To further understand the effects of the neurodegeneration, we analysed voluntary activity of the animals over the circadian cycle at 15 and 30 days of age (Fig. 3). At 15 days of age, no major differences between control and PPP knockdown animals can be seen (Fig. 3*A* and *B*–*E*). At 30 days of age, the morning activity of the animals is generally strongly reduced, leaving them with one activity peak per day, which is in accordance with published data (Luo et al., 2012; Fig. 3*A*). The maximum activity (evening activity peak) of PPP knockdown animals is reduced at this age compared to control animals (Fig. 3*A* and *H*). By contrast, the activity during rest phases is increased (Fig. 3*A*, *G* and *I*), although it is still low compared to the activity peak. Thus, progressive neurodegeneration leads to lower maximum activity and a loss of rhythmicity of the animal, as suggested by the increased activity during rest phases and reduced activity during activity phases (Fig. 3*G*, *H* and *I*). Thus, not only motor activity, but also the function of the clock neurons, that control circadian activity in *Drosophila*, appears to be progressively affected by neuronal PPP knockdown. Supporting this data, similar effects on the circadian rhythm have been reported previously upon systemic inhibition of the PPP (Rey et al., 2016). However, PPP inhibition in this report was drug-induced acute systemic inhibition in contrast to our chronic neuron-specific knockdown; thus, it is not clear whether the mechanisms that lead to a loss of rhythmicity are the same in both cases.

## Genetically encoded NADPH and H₂O₂ sensors reveal causal connection between lack of NADPH and high ROS

The oxidative branch of the PPP is considered to provide the major part of all cellular NADPH (Fan et al., 2014; Merritt et al., 2009; Stincone et al., 2015). To our knowledge, however, a functional link between NADPH reduction and concomitant ROS elevation has not been

established in living tissue. Moreover, experimental data linking glucose re-routing and oxidative stress is just available in (primary) cell culture (Kuehne et al., 2015; Moon et al., 2020). To measure NADPH in living tissues, we generated flies expressing the NADPH sensor iNap1 (Tao et al., 2017) under UAS-control, allowing for cell type specific expression. To assess functionality of the sensor, we expressed iNap1 in *Drosophila* eye imaginal discs (using eyeless-Gal4) and imaged *ex vivo* eye imaginal discs in buffers that interfere with cellular NADPH levels. CHP (100 μм) was used to induce oxidative stress and reduce NADPH levels, whereas NADPH addition (40 μM) was used to increase NADPH levels. As expected, addition of CHP reduces NADPH levels in the eye imaginal discs, whereas washing in NADPH elevates NADPH levels (Fig. 4*A*), indicating that iNap1 can be used reliably in living tissue to measure NADPH concentrations.

We then compared NADPH levels in control eye imaginal discs to eye imaginal discs with a neuronal PPP knockdown. Because sensors have different $K_D$s, measuring baseline fluorescence alone can hide fluctuations of the sensor's target, depending on how well the examined system is buffered (McMullen et al., 2022). In particular, the $K_D$ of iNap1 is very low, at 2 μм. Therefore, the sensor is probably close to saturation in a given tissue at all times. Hence, it is necessary to normalize a baseline fluorescence to relative fluorescence changes of the same sensor to gain more detailed information about the overall metabolic state of the examined tissue. Thus, we compared baseline values to peak (+40 μм NADPH) and minimum (+100 μм CHP) values to reveal potential NADPH fluctuations hidden in an otherwise saturated baseline fluorescence. In doing so, we see a significant reduction in cellular NADPH levels for *Pgd* knockdown, but not *G6Pdh* knockdown, albeit there is a clear trend here as well (Fig. 4; note: if the fluorescence values drop lower upon CHP application and recover less upon NADPH application in respect to the baseline, the ratio is greater. Thus, higher values indicate a reduction in homeostatic [NADPH]). Because pH-dependency of iNap1 fluorescence has been reported (Tao et al., 2017), we measured cellular pH in eye imaginal discs of the different genotypes used for the experiments and found no significant changes that could interfere with sensor function (Fig. 5).

Because NADPH is needed to reduce oxidized GSSG and replenish the GSH pool, it is indirectly involved in ROS clearance. To demonstrate a direct metabolic link between alternative glucose metabolism in neurons and ROS, we generated flies expressing the H₂O₂ sensor HyPerRed (Ermakova et al., 2014) under UAS-control. To test the sensor, we expressed HyPerRed in *Drosophila* eye imaginal discs. After measuring baseline H₂O₂-levels, H₂O₂ (600 μм) was added to ensure that the sensor has not been saturated under baseline conditions. After

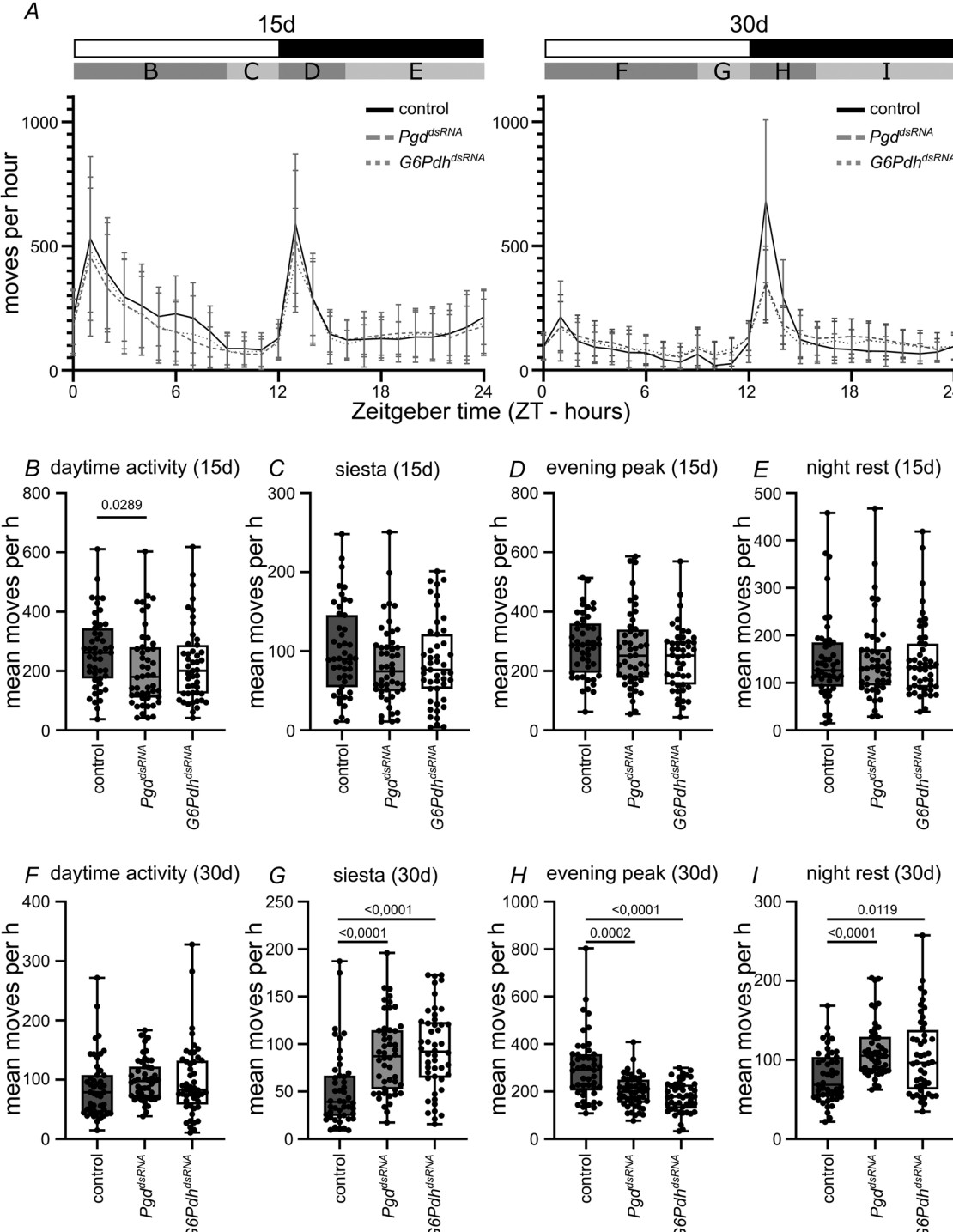

**Figure 3. Neuron-specific knockdown of PPP genes induces a progressive loss of maximum activity and circadian rhythmicity**

*A*, movements of flies over the circadian cycle for animals of the indicated genotypes and age (left 15 days, right 30 days). *B–E*, mean moves of 15 day-old animals for the indicated genotypes and time intervals. Daytime activity (ZT0-9; *B*), siesta (ZT9-12; *C*), evening peak (ZT12-16; *D*) and night rest (ZT16-24; *E*). *F–I*, mean moves of 30-day-old animals for the indicated genotypes and time intervals. Daytime activity (ZT0-9; *F*), siesta (ZT9-12; *G*), evening peak (ZT12-16; *H*) and night rest (ZT16-24; *I*). Statistically significant difference in total movements was determined using non-parametric one-way-ANOVA with Dunn's multiple comparison. Box plots: the box indicates the 25th and 75th percentile; the line within the box marks the median; whiskers represents minimum and maximum values. Each dot represents one fly. *P* values for significantly different conditions are shown; *P* values for non-significantly different conditions are not shown. dsRNA lines used: *GFP*^*dsRNA9331*^ (control), *G6Pdh*^*dsRNA50667*^, *Pgd*^*dsRNA65078*^. N = 3, n = 48.

$H_2O_2$ treatment pyruvate (10 mM) was added to scavenge $H_2O_2$. If the tissue survives $H_2O_2$ treatment, pyruvate treatment should reverse $H_2O_2$ levels (Fig. 6A–D). Unlike the $K_D$ of iNap1, the $K_D$ of HyPerRed is reported to be ∼30 μM, making saturation under baseline conditions improbable (Ermakova et al., 2014). Our experiments show that cellular $H_2O_2$ concentrations are well below saturation of the sensor because addition of $H_2O_2$ increases sensor fluorescence strongly (Fig. 6). Thus, fluctuations in cellular $H_2O_2$ levels can be seen in altered

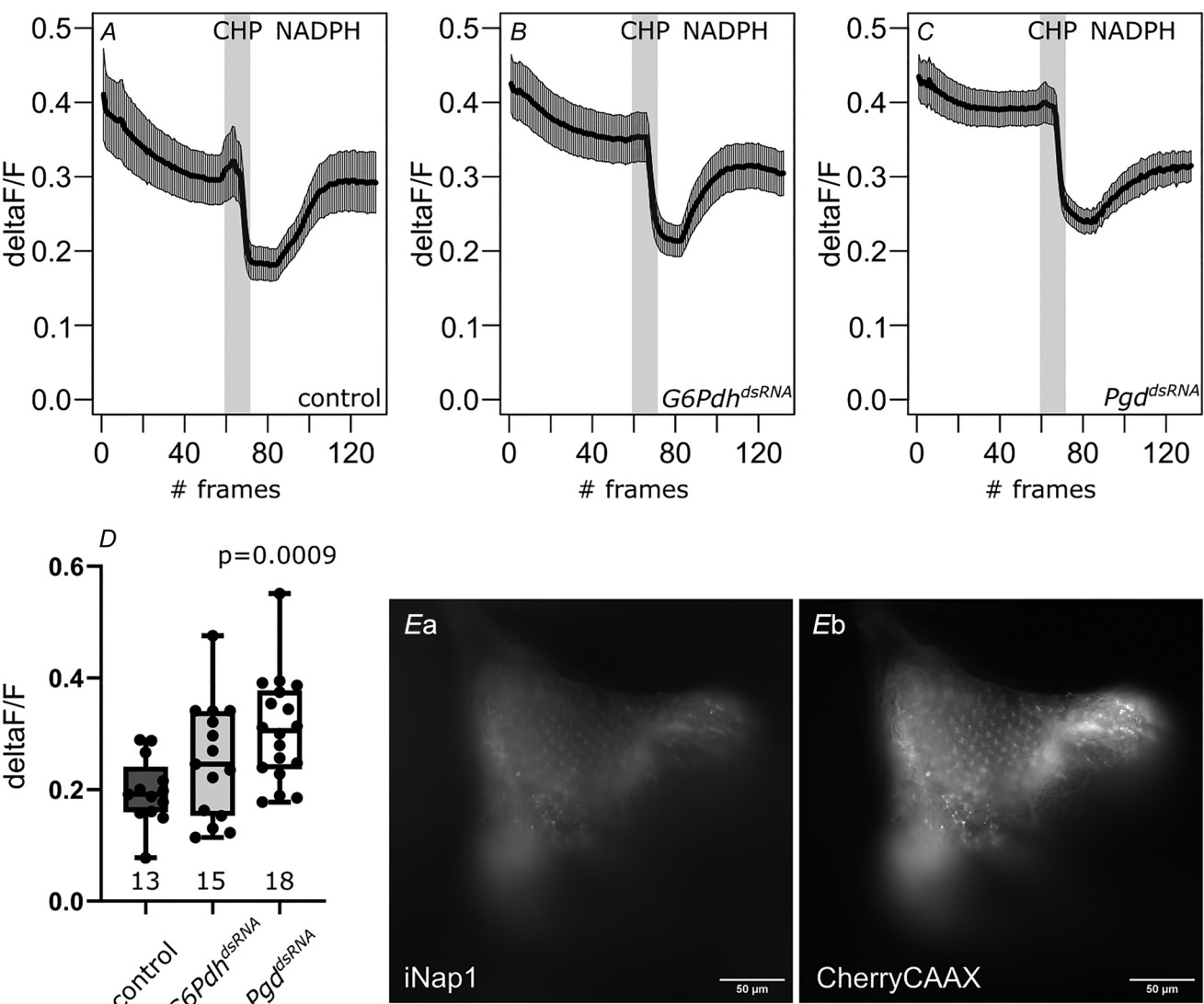

**Figure 4. PPP knockdown in eye imaginal discs reduces cellular NADPH levels**
Eye imaginal discs expressing the NADPH sensor iNap1, recombined to mCherryCAAX, show changes in fluorescence intensity of iNap1 (normalized to mCherryCAAX (NADPH insensitive) fluorescence) upon application of strong oxidants (CHP) or NADPH (A). NADPH baseline levels (normalized to peak and minimum; Note: if the fluorescence values drop lower upon CHP application and recover less upon NADPH application in respect to the baseline, the ratio is greater. Thus, higher values indicate a reduction in baseline NADPH levels.) in *G6Pdh* knockdown eye imaginal discs show a trend towards lower baseline levels (B and D). There is a significant reduction of NADPH baseline levels (normalized to peak and minimum) upon *Pgd* knockdown (C and D). Frame rate: 10 s; numbers under boxes represent the number of eye imaginal discs analysed. Whiskers extend to extremes; box width represents first and third quartile; black bar indicates median. Unpaired, two-tailed Mann–Whitney *U* test. *P* values for significantly different conditions are shown; *P* values for non-significantly different conditions are not shown. dsRNA lines used: *G6Pdh*[dsRNA101507]; *Pgd*[dsRNA100269]; *kl-3*[dsRNA32971] (control). E, representative image of a live eye imaginal disc expressing iNap (Ea) and mCherryCAAX (Eb).

baseline fluorescence. Indeed, compared with controls, the baseline fluorescence is significantly increased in neuronal *G6Pdh* knockdown eye imaginal discs, indicating elevated $H_2O_2$ levels (Fig. 6*B* and *D*). *Pgd* knockdown eye imaginal discs show a trend towards elevated $H_2O_2$ levels (Fig. 6*C* and *D*). This trend towards higher $H_2O_2$

levels was confirmed using a different dsRNA construct. Knockdown of *Pgd* using the second dsRNA construct resulted in significantly elevated $H_2O_2$ levels (Fig. 6*E*). These data indicate that a reduction of NADPH levels as a result of loss of PPP induces elevated $H_2O_2$ levels in *ex vivo Drosophila* eye imaginal disc neurons.

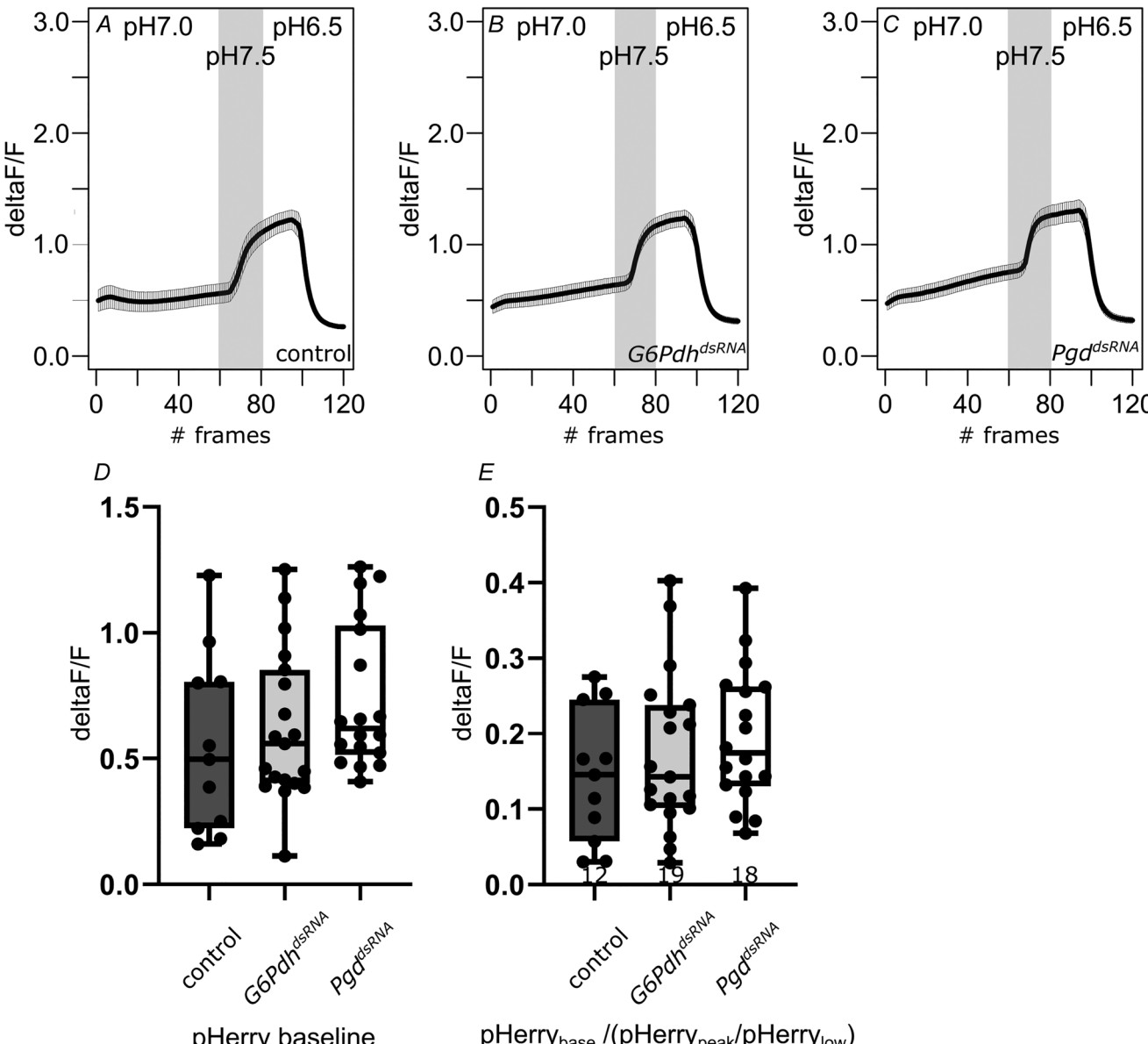

**Figure 5. oxPPP knockdown does not affect cellular pH**
Eye imaginal discs expressing the pH sensor pHerry were imaged to assess baseline cellular pH and the functionality of the sensor (*A–C*). The sensor reacts to pH changes as expected (*A*). There are no differences in baseline pH in *G6Pdh*$^{dsRNA\,101507}$ or *Pgd*$^{dsRNA\,100269}$ expressing eye imaginal discs compared to controls (*kl-3*$^{dsRNA\,32971}$), neither when comparing baseline fluorescence levels, nor when putting the baseline in relation to minimum and maximum fluorescence (*D* and *E*). Frame rate: 10 s; numbers under boxes represent the number of eye imaginal discs analysed. Whiskers extend to extremes; box width represents first and third quartile; black bar indicates median. Unpaired, two-tailed Mann–Whitney *U* test: no significant differences. *P* values for significantly different conditions are shown; *P* values for non-significantly different conditions are not shown.

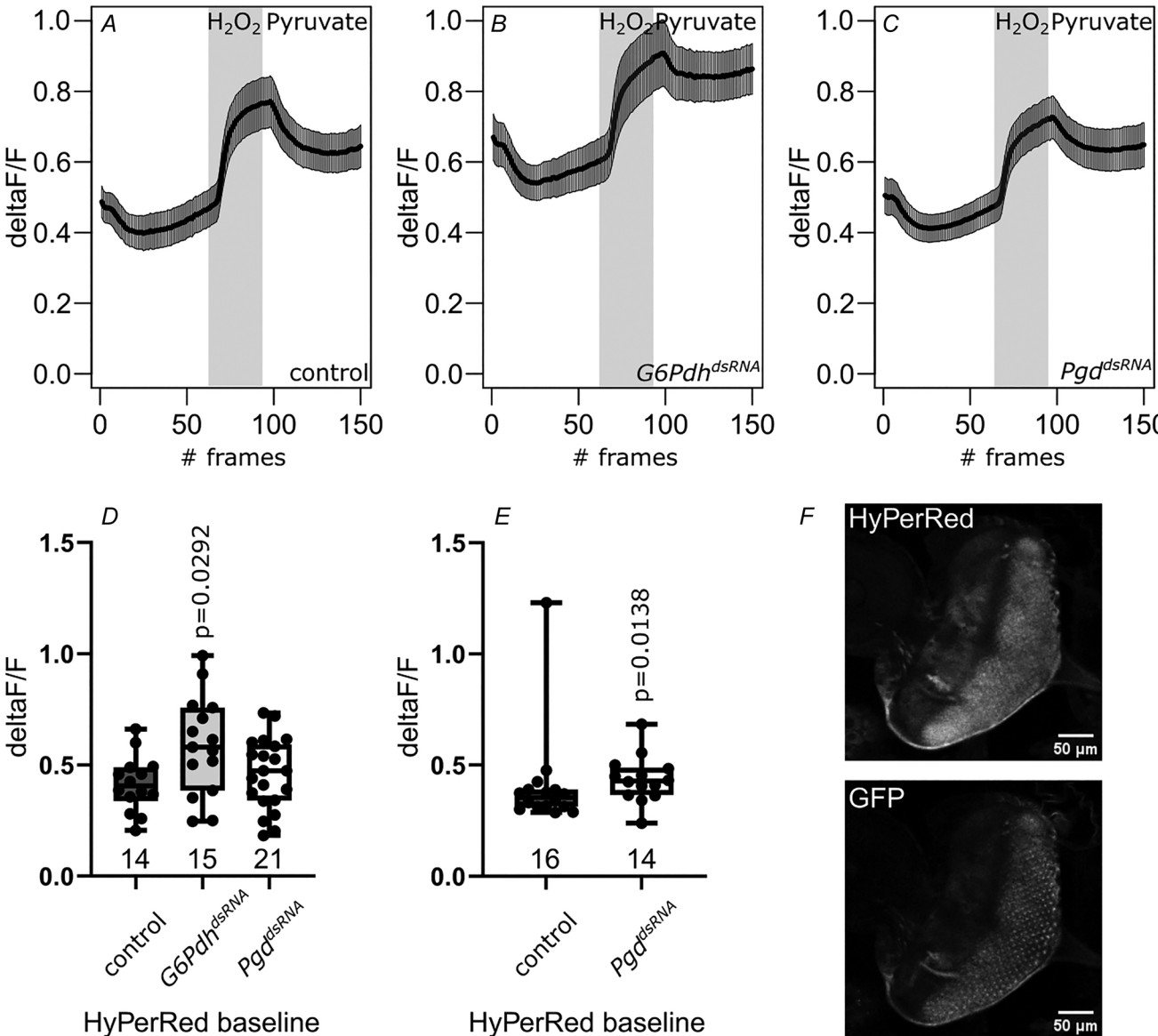

**Figure 6. HyPerRed imaging shows increased cellular $H_2O_2$ levels upon knockdown of PPP**

Eye imaginal discs expressing the $H_2O_2$ sensor HyPerRed, recombined to GFP were imaged *ex vivo*. HyPerRed is functional when expressed in *Drosophila* eye imaginal discs because changes in cellular $H_2O_2$ concentrations (induced by application of either 600 µM $H_2O_2$ or 10 mM pyruvate) are reflected in the expected changes in sensor fluorescence (*A*). Knockdown of *G6Pdh* (*B* and *D*) or *Pgd* (*C* and *D*) leads to increased basal $H_2O_2$ concentration in the cells of the eye imaginal disc. The change in baseline $H_2O_2$ levels in *G6Pdh* knockdowns is significant, whereas, in *Pgd* knockdown eye imaginal discs, a clear trend towards higher $H_2O_2$ levels can be seen. HyPer-Red fluorescence was normalized to GFP fluorescence ($H_2O_2$ insensitive). Frame rate: 10 s; numbers under boxes indicates the number of eye imaginal discs analysed. Whiskers extend to extremes; box width represents first and third quartile; black bar indicates median. Unpaired, two-sided Wilcoxon rank sum test, *P* = 0.03795; dsRNA lines used: *G6Pdh*$^{dsRNA\,101507}$; *Pgd*$^{dsRNA\,100269}$; *kl-3*$^{dsRNA\,32971}$ (control). *E*, knockdown of *Pgd* using *Pgd*$^{dsRNA\,65078}$ in neurons of the eye imaginal discs leads to a significant increase of basal $H_2O_2$ concentration in the cells, confirming the trend seen using *Pgd*$^{dsRNA\,100269}$ (*D*). Frame rate: 10 s; numbers under boxes indicate the number of eye imaginal discs analysed. Whiskers extend to extremes; box width represents first and third quartile; black bar indicates median. Unpaired, one-tailed Mann–Whitney *U* test. *P* values for significantly different conditions are shown; *P* values for non-significantly different conditions are not shown. *N* = 2; n 14–16. *F*, representative image of a live eye imaginal disc expressing HyPerRed (upper) and GFP (lower).

## Neuronal loss of PPP leads to a reduction of NADPH in neurons of the adult brain and induces elevated oxidative stress

We were able to individually show decreased NADPH levels and elevated $H_2O_2$ levels upon loss of PPP *ex vivo* in larval eye imaginal discs. However, eye imaginal discs are developing organs and thus the neurons there do not recapitulate all features of fully differentiated neurons in the adult brain. To analyse the function of the PPP in adult brain neurons, we used the above-described sensors to measure NADPH and $H_2O_2$ levels in neurons of *ex vivo* adult brains. To achieve this, we generated flies expressing both sensors, iNap1 and HyPerRed, simultaneously in all neurons. We then expressed dsRNA-constructs against either *G6Pdh* or *Pgd* together with the two sensors pan-neuronally in the adult animal, using the driver *elavGa4,tubGal80^{ts};elavGal4*. We dissected adult brains 5 days after dsRNA induction and imaged NADPH and $H_2O_2$ levels in the antennal lobes because these regions have a distinct recognizable pattern and are easily accessible for imaging (Fig. 7*A* and *B*).

There is a significant reduction of baseline NADPH levels in *Pgd* knockdown neurons, whereas *G6Pdh* knockdown does not induce significant changes (Fig. 7*A*). For *G6Pdh* knockdowns only a tendency towards lower NADPH levels can be seen (Fig. 7*A*). Interestingly, the reduction in NADPH levels is not accompanied by a change in $H_2O_2$ levels (Fig. 7*B*). This is probably because of the young age of the animals that might not give sufficient time for GSH pool depletion and $H_2O_2$ accumulation. Unfortunately, neither imaging of iNap1, nor HyPerRed was possible in older knockdown animals, since sensor fluorescence was completely lost (data not shown). Interestingly, loss of sensor fluorescence was not as strong in aged control animals, indicating that a redox disbalance interferes with sensor functionality. To assess oxidative damage in older animals, we analysed the amount of oxidized lipids (using 4-HNE staining) in 20-day-old animals (Fig. 7*C*). Knockdown of *G6Pdh* induces elevated levels of oxidized lipids, indicating elevated ROS stress, whereas *Pgd* knockdown does not induce this phenotype indicating differences in *G6Pdh* and *Pgd* knockdown phenotypes. In summary, our data suggest the PPP is needed to maintain stable NADPH levels and ROS balance in fully developed neurons.

## Discussion

In an ageing society, the prevalence of neurodegenerative diseases will increase to be a major contributing factor to overall population health and will challenge the robustness of welfare systems. A role of neuronal ROS imbalance has been suggested as a factor in the development or progression of multiple neuro-degenerative diseases (Sienes Bailo et al., 2022). It is widely accepted that PPP-dependent production of NADPH in the neurons is essential to cope with ROS stress (Bonvento & Bolaños, 2021). And yet, up to now, there has been no confirmation of a functional link between the activity of the PPP and ROS levels in living nervous tissue. Here, we show that neuronal loss of PPP in the adult fly induces neurodegeneration and behavioural phenotypes (Figs 2 and 3). The phenotype can be described as mild compared to other neurodegenerative phenotypes reported previously (Melentev et al., 2021). One reason for this could be that the PPP is not the only source of NADPH in the cell. There are also other enzymes that contribute to the pool, such as malic enzyme or isocitrate dehydrogenase (Chen et al., 2019; Fan et al., 2014). Interestingly, malic enzyme and isocitrate dehydrogenase have been shown to be upregulated in cells with a *G6Pdh* deletion (Chen et al., 2019), a compensatory effect that reduces the effect of G6Pdh loss and could also be in place in the model studied here. The neurodegenerative phenotype we see can be rescued by food-supplementation with antioxidants, indicating a link between PPP inhibition and ROS stress (Fig. 2). Interestingly, elevated ROS levels have been observed in ageing animals and overexpression of *G6Pdh* increases life span in *Drosophila* and mice arguing for a general role of ROS in ageing (Fei et al., 2025; Legan et al., 2008; Nóbrega-Pereira et al., 2016; Scialò et al., 2016). Progressive neurodegeneration due to elevated ROS is accompanied by neuronal dysfunction (Figs 2 and 3). It has been shown recently that neuronal glucose uptake via Glut1 and neuronal PPP is also essential for complex neuron-mediated capabilities of the animal, such as learning and memory (de Tredern et al., 2021). On the other hand, mildly elevated ROS appear to strengthen synaptic connections and thus improve motor function (Oswald et al., 2018), indicating that ROS might have different effects on neuronal function depending on their concentration.

Although such analyses argue for a causal link between PPP and ROS disbalance, they do not show a functional link on the molecular level. Thus, we turned to metabolite live imaging to determine whether down-regulation of PPP indeed decreases cellular NADPH levels and increases $H_2O_2$ levels. We expressed iNap1 and/or HyPer-Red in *Drosophila* eye imaginal discs and adult brains. Indeed, inhibition of PPP via knockdown of *Pgd* or *G6Pdh* in larval eye imaginal discs and adult neurons altered cellular NADPH levels (Figs 4 and 7). Interestingly, baseline NADPH concentrations appear to differ between the developmental stages because iNap1 appears to be saturated at baseline levels in eye imaginal discs even upon PPP knockdown. By contrast, a reduction of NADPH levels in adult neurons can be detected using the same sensor (Figs 4 and 7). In accordance with the hypothesis that cellular NADPH is used to maintain intracellular

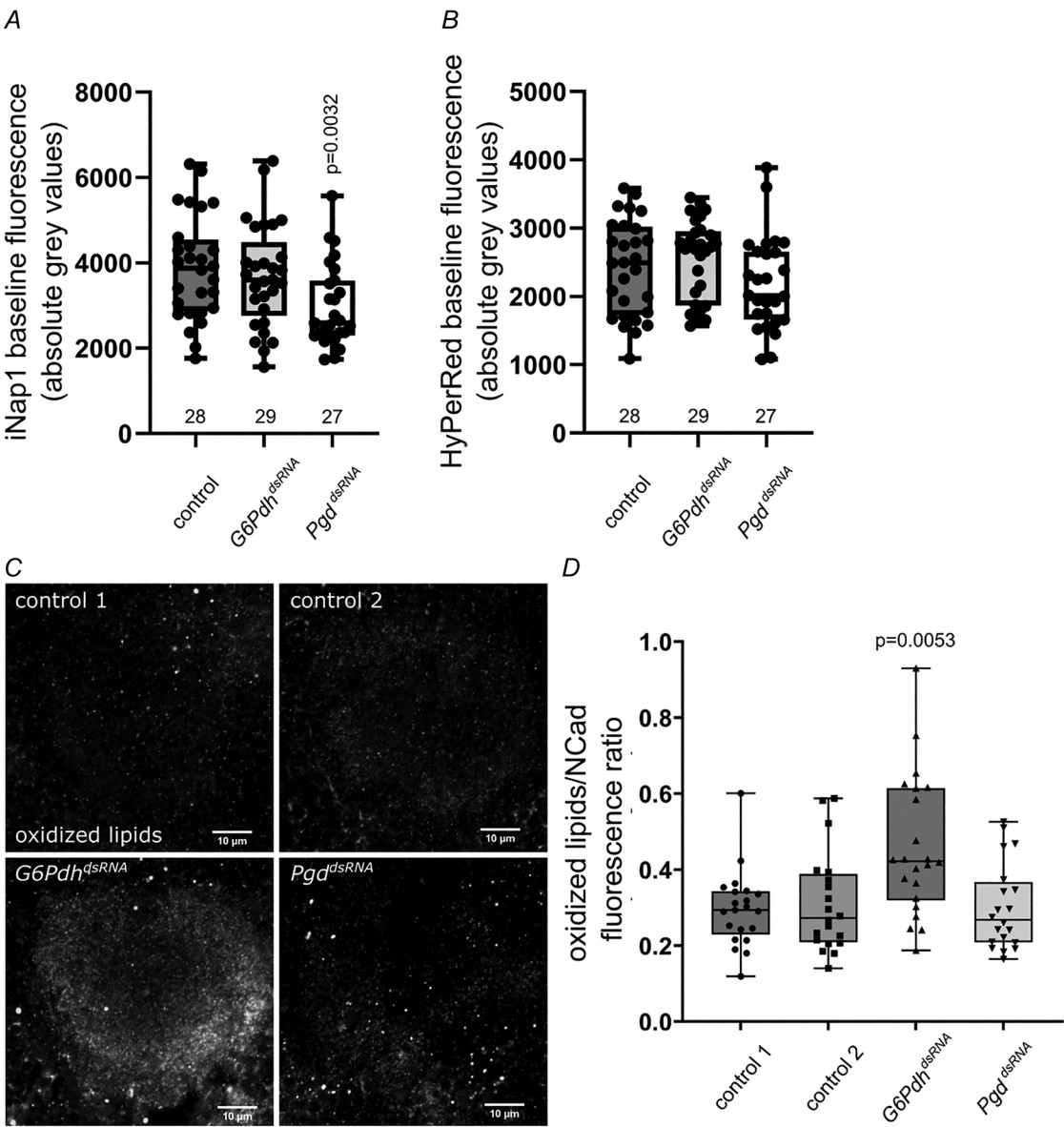

**Figure 7. Multiparametric imaging of both sensors in *ex vivo* adult brains reveals reduced NADPH levels upon loss of PPP**

*A* and *B*, adult brains of 5-day-old animals that pan-neuronally express both sensors, iNap1 (*A*) and HyPerRed (*B*), were imaged *ex vivo*. Fluorescence was measured in the antennal lobes. Loss of PPP reduces NADPH levels in neurons. Compared wth controls (*kl-3^dsRNA32971^*), the reduction is significant for *Pgd* knockdown (*Pgd^dsRNA65078^*), whereas only a trend can be seen in *G6Pdh* knockdown neurons (*G6Pdh^dsRNA3337^*). Interestingly, $H_2O_2$ levels remain unaltered. Numbers under boxes represent the number of antennal lobes analysed. Whiskers extend to extremes; box width represents first and third quartile; Black bars indicate median; Unpaired, two-sided Wilcox rank sum test. *C*, to assess ROS-induced damage oxidized lipids have been stained using $\alpha$ 4-HNE in brains of 20-day-old female flies. Shown is oxidized lipid staining in the mushroom body calyx. Knockdown of *G6Pdh* induces elevated levels of oxidized lipids. dsRNA lines used: *GFP^dsRNA9331^* (control 1), *kl-3^dsRNA32971^* (control, control 2), *G6Pdh^dsRNA50667^*, *Pgd^dsRNA65078^*. $N = 3$, $n = 12$. *D*, quantification of oxidized lipids (C). Fluorescence intensity was measured in the mushroom body calyx and normalized to NCad staining. dsRNA lines used: *GFP^dsRNA9331^* (control 1), *kl-3^dsRNA32971^* (control, control 2), *G6Pdh^dsRNA50667^*, *Pgd^dsRNA65078^*. $N = 3$, $n = 12$. Whiskers extend to extremes; box width represents first and third quartile; Black bars indicate median; Unpaired, two-sided Wilcoxon Rank Sum Test. Kruskal–Wallis with Dunn's test for multiple comparison. *P* values for significantly different conditions are shown; *P* values for non-significantly different conditions are not shown.

ROS homeostasis, knockdown of PPP leads to increased $H_2O_2$ levels in *ex vivo* eye imaginal discs (Fig. 6). This clearly shows that there is a functional link between PPP, NADPH production and ROS scavenging in neuronal cells in *Drosophila* in living tissue. Unfortunately, live imaging of NADPH and $H_2O_2$ levels in neurons of the adult brain was only possible in young animals. Here, cellular NADPH levels are reduced upon loss of PPP (Fig. 7); however, a concomitant rise in $H_2O_2$ levels could not be seen. The very short time (5 days) of PPP knockdown induction could be the reason for that. Most probably, this time frame is not sufficient to deplete the GSH pool and increase $H_2O_2$ levels. Interestingly, it was not possible to image either sensor in older animals because sensor fluorescence was lost in PPP knockdown neurons (not in control knockdown neurons), indicating that sensor function is impaired by prolonged NADPH-/ROS-imbalance. Using an increase in the amount of oxidized lipids as a readout for elevated ROS stress, we were able to confirm the predicted ROS imbalance.

In the present study, we provide the first direct evidence that the PPP is a major contributor to cellular NADPH levels in living tissue. Furthermore, we show that a reduction of cellular NADPH levels is accompanied by an increase in cellular $H_2O_2$ levels and thus ROS imbalance. We further highlight the importance of the neuronal PPP and its effect on ROS homeostasis for healthy ageing.

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

## Additional information

### Data availability statement

All raw data are available from the corresponding author upon reasonable request.

### Competing interests

The authors declare that they have no competing interests.

### Author contributions

All experiments have been performed at the Schirmeier lab at WWU Münster (until mid-2021) and TU Dresden (from mid-2021). S.M., N.S., A.K.S., I.N., A.F. and S.G. were responsible for investigations. S.M. was responsible for methodology. S.M., N.S. and A.K.S. were responsible for formal analysis. S.M., N.S., A.K.S., I.N., A.F. and S.G. were responsible for validation. S.M., N.S., A.K.S. and S.S. were responsible for visualization. S.M. and S.S. were responsible for writing the original draft. S.S. was responsible for conceptualization, funding acquisition, data curation, supervision, and reviewing and editing. All authors approved the final version of the manuscript submitted for publication and agree to be accountable for all aspects of the work in ensuring that questions related to the accuracy or integrity of any part of the work are appropriately investigated and resolved. All persons listed as authors qualify for authorship.

### Funding

This work was supported by funding from the DFG to SS (SFB 1009, SCHI 1380/6-1).

### Acknowledgements

## Keywords

neurodegeneration, NADPH, oxidative stress, pentose phosphate pathway, reactive oxygen species

## Supporting information

Additional supporting information can be found online in the Supporting Information section at the end of the HTML view of the article. Supporting information files available:

**Peer Review History**

