## [Peer Review History · The Journal of Physiology]

Neuronal loss of the Pentose Phosphate Pathway in the living nervous system is causally linked to [NADPH] reduction and elevated oxidative stress

Stephan Müller, Nina Surina, Andrés Köhler-Solís, Ioannis Nellas, Astrid Fleige, Sebastian Görtz, and Stefanie Schirmeier
DOI: 10.1113/JP288582

Corresponding author(s): Stefanie Schirmeier (stefanie.schirmeier@tu-dresden.de)

Review Timeline:

Submission Date:	20-Jan-2025
Editorial Decision:	11-Apr-2025
Revision Received:	09-Jan-2026
Accepted:	18-Feb-2026

Senior Editor: Katalin Toth

Reviewing Editor: Valentina Mosienko

Transaction Report:

Dear Dr Schirmeier,

Re: JP-RP-2025-288582 **"Neuronal loss of the Pentose Phosphate Pathway in the living nervous system is causally linked to [NADPH] reduction and elevated oxidative stress"** by Stephan Müller, Nina Surina, Andrés Köhler-Solís, Ioannis Nellas, Astrid Fleige, Sebastian Görtz, and Stefanie Schirmeier

Thank you for submitting your manuscript to The Journal of Physiology. It has been assessed by a Reviewing Editor and by 1 expert referee and we are pleased to tell you that it is potentially acceptable for publication following satisfactory major revision.

REVISION CHECKLIST:

We look forward to receiving your revised submission.

Yours sincerely,

Katalin Toth
Senior Editor
The Journal of Physiology

REQUIRED ITEMS

- Author photo and profile. First or joint first authors are asked to provide a short biography (no more than 100 words for one author or 150 words in total for joint first authors) and a portrait photograph. These should be uploaded and clearly labelled together in a Word document with the revised version of the manuscript. See Information for Authors for further details.

- Your manuscript must include a complete Additional Information section, including competing interests; funding; author contributions and acknowledgements.

- Please upload separate high-quality figure files via the submission form.

- Please ensure that any tables are editable and in Word format, and wherever possible, embedded in the article file itself.

- Please ensure that the Article File you upload is a Word file.

- Your paper contains Supporting Information of a type that we no longer publish, including supplementary tables and figures. Any information essential to an understanding of the paper must be included as part of the main manuscript and figures. The only Supporting Information that we publish are video and audio, 3D structures, program codes and large data files. Your revised paper will be returned to you if it does not adhere to our Supporting Information Guidelines.

- Papers must comply with the Statistics Policy: https://jp.msubmit.net/cgi-bin/main.plex?form_type=display_requirements#statistics.

In summary:

- If $n \leq 30$, all data points must be plotted in the figure in a way that reveals their range and distribution. A bar graph with data points overlaid, a box and whisker plot or a violin plot (preferably with data points included) are acceptable formats.

- If $n > 30$, then the entire raw dataset must be made available either as supporting information, or hosted on a not-for-profit repository, e.g. FigShare, with access details provided in the manuscript.

- 'n' clearly defined (e.g. x cells from y slices in z animals) in the Methods. Authors should be mindful of pseudoreplication.
- All relevant 'n' values must be clearly stated in the main text, figures and tables.
- The most appropriate summary statistic (e.g. mean or median and standard deviation) must be used. Standard Error of the Mean (SEM) alone is not permitted.
- Exact p values must be stated. Authors must not use 'greater than' or 'less than'. Exact p values must be stated to three significant figures even when 'no statistical significance' is claimed.

- Please include an Abstract Figure file, as well as the Figure Legend text within the main article file. The Abstract Figure is a piece of artwork designed to give readers an immediate understanding of the research and should summarise the main conclusions. If possible, the image should be easily 'readable' from left to right or top to bottom. It should show the physiological relevance of the manuscript so readers can assess the importance and content of its findings. Abstract Figures should not merely recapitulate other figures in the manuscript. Please try to keep the diagram as simple as possible and without superfluous information that may distract from the main conclusion(s). Abstract Figures must be provided by authors no later than the revised manuscript stage and should be uploaded as a separate file during online submission labelled as File Type 'Abstract Figure'. Please also ensure that you include the figure legend in the main article file. All Abstract Figures should be created using BioRender. Authors should use The Journal's premium BioRender account to export high-resolution images. Details on how to use and access the premium account are included as part of this email.

EDITOR COMMENTS

Reviewing Editor:

Comments for Authors to ensure the paper complies with the Statistics Policy (Required):
Please, make sure that all graphs include individual data points. If possible, please include all figures in the main part of the manuscript.

Comments to the Author:
Many thanks for submitting your work to the Journal.

The reviewer highlighted several areas of improvement, these include the need to perform additional experiments in order to strengthen the authors' conclusions. In the current format, there are too many open questions. Constructive comments from the reviewer will help the revision to ease these concerns. The Journal of Physiology does not support supplemental figures, please include these into the main text.

Please also see 'Required Items' above.

REFEREE COMMENTS

Referee #2:

This manuscript by Müller et al. presents compelling evidence supporting the hypothesis that neuronal glucose uptake and its metabolism via the pentose phosphate pathway (PPP) are crucial for NADPH production, counteracting the high oxidative stress inherent to neuronal metabolism. The authors' findings suggest that: (a) PPP activity is required for neuronal maintenance during aging; and (b) PPP knockdown leads to reduced neuronal NADPH levels and increased H₂O₂ in adult optic lobes and/or ex vivo larval tissues, as measured using novel NADPH and H₂O₂ biosensors. This manuscript brings valuable insights into PPP function in vivo and provides important new tools for investigating the roles of NADPH and ROS in various cellular processes across different organs.

However, I have several points that I believe would strengthen the manuscript and its conclusions:

1 - The authors focus on knocking down Pgd and Zw (PPP enzymes) to disrupt NADPH synthesis. Given previous work from the lab suggesting that glial cells supply glucose to neurons via trehalose breakdown, it would be insightful to explore

whether glial hexokinase knockdown or neuronal Glut1 knockdown (blocking glucose transport) phenocopies the effects of PPP inhibition. While this might be beyond the current scope, it could provide a broader perspective on how glia-neuron metabolic exchange fuels diverse metabolic pathways. Even a brief discussion of this potential interplay would be valuable.

2 - While the authors demonstrate a significant contribution of PPP to the NADPH pool, which is important for ROS regulation, it's important to acknowledge that other enzymes (malic enzyme, isocitrate dehydrogenase, and glutamate dehydrogenase) also contribute to NADPH production and their potential role in the neuronal ROS regulation, as their role has not been formally tested.

3 - Regarding Figure 1, depicting brain changes with age, I suggest incorporating a cell death assay (e.g., TUNEL or a marker for apoptosis) in addition to the "holes" measurement. This might reveal earlier stages of cell death and provide a more sensitive measure of neurodegeneration. The current "holes" measurement likely reflects later stages where debris has been cleared. Furthermore, while it might be challenging to include, linking these neurological defects to a behavioral output (e.g., a climbing assay) and assessing rescue with antioxidants would significantly strengthen the findings. If this cellular phenotype has already been correlated with such behavioral changes, a reference should be included.

4 - The experiments in Figure 1 (panels C-D) demonstrate that the PPP knockdown phenotype is likely dependent on oxidative stress, but they don't definitively show that ameliorating oxidative stress specifically in neurons improves the phenotype. Systemic antioxidant treatment could have broader effects. To address this, experiments increasing NADPH levels specifically in neurons (via pathways independent of PPP) would be necessary to confirm the neuron-autonomous effect of NADPH on neurodegeneration.

5 - While I understand the technical challenges with sensor measurements in older knockdown animals, it's important to show how H₂O₂ and NADPH levels change in wild-type brains during aging. This would contextualize the data in Figure 1 and indicate whether the capacity to buffer oxidative stress changes with age. This is important for correlating the observed structural changes with the redox state.

6 - For Figures 2 and 3, please include representative images underlying the sensor measurements. This will allow readers to assess the quality of the data. While measurements from whole optic lobes are relevant to Figure 1, measuring the signal in the whole brain might reveal regional differences in oxidative stress and NADPH levels, potentially identifying areas more susceptible to ROS and more dependent on NADPH. Has this been assessed?

7 - Please clarify in the figure legends whether p-values are not shown for non-significant conditions. The presentation of p-values and sample sizes (n) is inconsistent across figures. Ensure consistency in both placement (on graphs or in legends) and the reporting of n values.

8 - In Supplementary Figure 1, data points should be shown.

9 - The description of the activity-dependent neurodegeneration experiments in the Methods section ("For activity dependent neurodegeneration experiments, female, mated flies were either kept in constant light or constant darkness.") needs further clarification. What constitutes "activity-dependent neurodegeneration," and why are constant light and darkness used? Please provide more detail about the rationale of the experimental setup.

END OF COMMENTS

We thank the editor and the reviewer for their constructive criticism. We did additional experiments and adapted the manuscript text, including the discussion, according to the comments. We are convinced that the manuscript is significantly improved now. Please find a point-by-point reply below. Our answers are highlighted in blue. We further provide one version of the revised manuscript in which the changes we made are highlighted in blue.

EDITOR COMMENTS

Reviewing Editor:

Comments for Authors to ensure the paper complies with the Statistics Policy (Required): Please, make sure that all graphs include individual data points. If possible, please include all figures in the main part of the manuscript.

We changed the figures and their order accordingly. All figures now include individual data points. All figures are main figures now.

Comments to the Author:

Many thanks for submitting your work to the Journal.

The reviewer highlighted several areas of improvement, these include the need to perform additional experiments in order to strengthen the authors' conclusions. In the current format, there are too many open questions. Constructive comments from the reviewer will help the revision to ease these concerns. The Journal of Physiology does not support supplemental figures, please include these into the main text.

All figures are main figures now.

Please also see 'Required Items' above.

Referee #2:

This manuscript by Müller et al. presents compelling evidence supporting the hypothesis that neuronal glucose uptake and its metabolism via the pentose phosphate pathway (PPP) are crucial for NADPH production, counteracting the high oxidative stress inherent to neuronal metabolism. The authors' findings suggest that: (a) PPP activity is required for neuronal maintenance during aging; and (b) PPP knockdown leads to reduced neuronal NADPH levels and increased H₂O₂ in adult optic lobes and/or ex vivo larval tissues, as measured using novel NADPH and H₂O₂ biosensors. This manuscript brings valuable insights into PPP function in vivo and provides important new tools for investigating the roles of NADPH and ROS in various cellular processes across different organs.

However, I have several points that I believe would strengthen the manuscript and its conclusions:

1 - The authors focus on knocking down Pgd and Zw (PPP enzymes) to disrupt NADPH synthesis. Given previous work from the lab suggesting that glial cells supply glucose to neurons via trehalose

breakdown, it would be insightful to explore whether glial hexokinase knockdown or neuronal Glut1 knockdown (blocking glucose transport) phenocopies the effects of PPP inhibition. While this might be beyond the current scope, it could provide a broader perspective on how glia-neuron metabolic exchange fuels diverse metabolic pathways. Even a brief discussion of this potential interplay would be valuable.

These manipulations have been attempted in ours and other labs previously. Knockdowns of hexokinase are challenging, since the *Drosophila* genome encodes three hexokinases. Two of them seem to be broadly expressed, while one is reported to be testis-specific. Knockdown of either one of the broadly expressed hexokinases does not give a phenotype in most tissues, including glial cells. Thus, I assume their functions are redundant. Since glial knockdown of other glycolytic enzymes is lethal, any effects on neuronal PPP would likely be masked by the loss of glial glycolysis (even if the knockdown is restricted to adulthood, the animals die rather fast and neurodegeneration sets in early, see Volkenhoff *et al*, 2015).

Concerning knockdowns of neuronal Glut1, it has been shown that mushroom body neuron-specific Glut1 knockdown inhibits the formation of long-term memory in *Drosophila* (de Tredern *et al*, 2021). The study suggests that glucose transported via Glut1 is fueled into the PPP in this setting. Which is in agreement with our data. However, a direct effect on NADPH or H₂O₂ levels has not been shown in this study, thus the effects on memory formation could also be due to an inhibition of neuronal plasticity due to a lack of other metabolites generated in the PPP. We included this point in the discussion.

2 - While the authors demonstrate a significant contribution of PPP to the NADPH pool, which is important for ROS regulation, it's important to acknowledge that other enzymes (malic enzyme, isocitrate dehydrogenase, and glutamate dehydrogenase) also contribute to NADPH production and their potential role in the neuronal ROS regulation, as their role has not been formally tested. Thank you for mentioning this. We are aware and added the information on other sources of NADPH and the possibility of compensation to the discussion.

3 - Regarding Figure 1, depicting brain changes with age, I suggest incorporating a cell death assay (e.g., TUNEL or a marker for apoptosis) in addition to the "holes" measurement. This might reveal earlier stages of cell death and provide a more sensitive measure of neurodegeneration. The current "holes" measurement likely reflects later stages where debris has been cleared. Furthermore, while it might be challenging to include, linking these neurological defects to a behavioral output (e.g., a climbing assay) and assessing rescue with antioxidants would significantly strengthen the findings. If this cellular phenotype has already been correlated with such behavioral changes, a reference should be included.

Thank you for this comment. In the past we have correlated the neurodegenerative "hole" phenotype with apoptosis (Volkenhoff *et al*, 2015). Unfortunately, the antibody that we had used against activated Caspase3 does not exist anymore. The supplier has changed the way the antibody is produced, which led to the loss of recognition of *Drosophila* act. Casp3. Thus, we now attempted to do TUNEL staining. Unfortunately, the assay does not work properly in our hands. Even though we have used published protocols, we always detect strong unspecific staining in all nuclei in the surface regions of the brain (see Reviewer Figure 1). Also here, it seems that the supplier recently changed the composition of the kit.

Reviewer Figure 1: TUNEL assay yields unspecific staining. Shown is a confocal section of one hemisphere of a brain of a 30d old control animal (genotype: *w*; *elav-Gal4/+*; *elav-Gal4/UAS-GFPdsRNA*) stained for DAPI (cyan) and TUNEL (red). At the circumference of the brain, all nuclei are positive for TUNEL. From published data, one expects only few cells dispersed throughout the brain in the control samples. Since all samples of all genotypes showed the same TUNEL pattern, we are very sure that the staining is unspecific.

The neurodegenerative “hole” phenotype has been associated with progressive loss of climbing ability before (Sunderhaus *et al*, 2019; Cabirol-Pol *et al*, 2017; Hindle *et al*, 2017). To extend the data to another readout, we employed a different behavioral assay. We analyzed voluntary activity of the animals over the circadian cycle. The rest and activity periods are controlled by the circadian neurons of the CNS. We see a generally reduced activity during the morning in aged compared to young animals, which is in accordance with published results (Luo *et al*, 2012). In PPP KD animals, the evening activity peak of the animals is reduced at 30d of age, while we see heightened activity during rest periods. Both effects are absent in younger KD animals. This indicates, that on the one hand the maximum activity of the KD animals declines over time and on the other hand, the KD animals lose rhythmicity, since the difference between activity and rest phases is reduced. Thus, not only the maximum activity of the animals, but also the function of the circadian circuit is compromised with age in the PPP-KD animals. We included these findings as new Figure 3. Unfortunately, our attempts to rescue the phenotype with antioxidant food failed. This is surprising, since we were able to rescue the “hole” phenotype and it has been shown previously that paraquat-induced loss of circadian rhythmicity can be rescued using food supplemented with antioxidants (Vrailas-Mortimer *et al*, 2012). Although in this publication plant extracts have been used and thus, it cannot be excluded that other components than antioxidants conveyed the rescue. The problem in our case is likely the concentration of antioxidants used. It has been shown previously that each antioxidant works in a concentration-dependent manner (Ghanty *et al*, 2025; Yang *et al*, 2025). In addition, different neuronal types might have differential access to food-derived antioxidants. Unfortunately, we were not able to calibrate the antioxidant concentration needed to rescue the decline in circadian rhythmicity, since we would have needed a vast number of animals and measurements for this. We did not include the antioxidant food data into the manuscript, but can of course do so if the reviewer and/or the editor would prefer it.

4 - The experiments in Figure 1 (panels C-D) demonstrate that the PPP knockdown phenotype is likely dependent on oxidative stress, but they don't definitively show that ameliorating oxidative stress specifically in neurons improves the phenotype. Systemic antioxidant treatment could have broader effects. To address this, experiments increasing NADPH levels specifically in neurons (via pathways independent of PPP) would be necessary to confirm the neuron-autonomous effect of NADPH on neurodegeneration.

Thank you for this input. We agree that we cannot completely rule out a non-cell autonomous effect of the antioxidant food. However, since we knock down G6Pdh or Pgd specifically in neurons and also measure the changes in NADPH and H₂O₂ levels specifically in neurons, we know that a reduction of NADPH levels and an increase of oxidative stress occurs there. Since all other cell types are wild type and do not express the construct inducing the knockdowns of either G6pdh or Pgd, we assume that NADPH levels in those cells should be normal. Thus, we think that it is very likely that the reduction of NADPH levels specifically in neurons directly causes ROS increase in neurons, which can be rescued by systemic treatment with antioxidants.

Nonetheless, we thought of strategies to achieve a neuron specific rescue of the ROS stress. Overexpression of malic enzyme should increase NADPH levels and reduce oxidative stress, however, does not seem promising, since it has been shown that overexpression of malic enzyme counterintuitively can increase ROS in some tissues (Kim *et al*, 2015). Thus, interpretation of the phenotype of knockdown of PPP combined with overexpression of malic enzyme would be very difficult. As for overexpression of Isocitrate dehydrogenase or glutamate dehydrogenase, the expression level of these enzymes has effects on neuronal activity and energy metabolism. Both could lead to effects on neuronal survival by themselves (Ugur *et al*, 2017; Hohnholt *et al*, 2018). Thus, also here, an interpretation of the results would be difficult. Thus, we did not attempt these experiments.

5 - While I understand the technical challenges with sensor measurements in older knockdown animals, it's important to show how H₂O₂ and NADPH levels change in wild-type brains during aging. This would contextualize the data in Figure 1 and indicate whether the capacity to buffer oxidative stress changes with age. This is important for correlating the observed structural changes with the redox state.

Thank you for this comment. It has been shown in different studies that ROS levels increase with age in the brain of *Drosophila* (Fei *et al*, 2025; Scialò *et al*, 2016; reviewed in Lennicke & Cochemé, 2020). Our data supports these studies and suggests that elevated ROS levels and the accompanying shift to a more oxidizing redox state of the cell contribute to aging. We added this information to the discussion.

6 - For Figures 2 and 3, please include representative images underlying the sensor measurements. This will allow readers to assess the quality of the data. While measurements from whole optic lobes are relevant to Figure 1, measuring the signal in the whole brain might reveal regional differences in oxidative stress and NADPH levels, potentially identifying areas more susceptible to ROS and more dependent on NADPH. Has this been assessed?

In figures 2 and 3 (now 4 and 6) the effect of PPP knockdown of NADPH or H₂O₂ concentrations was measured in eye imaginal discs. We now included representative images from the live imaging experiments in both figures. Since the expression of the respective sensors and knockdowns was restricted to the neurons, we imaged the whole imaginal disc and used the signal from the whole imaginal disc for analysis. We did not observe any regional differences here.

This might be different when looking at the adult brain. However, here (Fig. 7), we just imaged the antennal lobes and thus cannot assess regional differences.

7 - Please clarify in the figure legends whether p-values are not shown for non-significant conditions. The presentation of p-values and sample sizes (n) is inconsistent across figures. Ensure consistency in both placement (on graphs or in legends) and the reporting of n values.

Thank you for this comment. We modified the figure legends accordingly.

8 - In Supplementary Figure 1, data points should be shown.

Thank you for pointing this out. We are sorry for not including the data points originally. We corrected this mistake. We also found that the N we had given was wrong, we corrected this as well. We apologize for this oversight. We checked all other N- and n-numbers in the manuscript and found no further mistakes.

9 - The description of the activity-dependent neurodegeneration experiments in the Methods section ("For activity dependent neurodegeneration experiments, female, mated flies were either kept in constant light or constant darkness.") needs further clarification. What constitutes "activity-dependent neurodegeneration," and why are constant light and darkness used? Please provide more detail about the rationale of the experimental setup.

Thank you for pointing this out. We are very sorry for this mistake. We did not perform any experiments in constant light or constant darkness. This sentence should not have been included in the methods section. We removed the sentence.

References:

- Cabirol-Pol M-J, Khalil B, Rival T, Faivre-Sarrailh C & Besson MT (2017) Glial lipid droplets and neurodegeneration in a *Drosophila* model of complex I deficiency. *Glia*: n/a--n/a
- Fei L, Liang Y, Kintscher U & Sigrist SJ (2025) Coupling of mitochondrial state with active zone plasticity in early brain aging. *Redox Biol* 79: 103454
- Ghanty S, Ganguly A, Nanda S, Mandi M, Das K, Biswas G, Maitra P, Khatun N & Rajak P (2025) Antioxidant and Pro-oxidant properties of naringenin: Unveiling the biphasic impacts on model, *Drosophila melanogaster*. *Food Chem Adv* 9: 101137
- Hindle SJ, Hebbar S, Schwudke D, Elliott CJH & Sweeney ST (2017) A saposin deficiency model in *Drosophila*: Lysosomal storage, progressive neurodegeneration and sensory physiological decline. *Neurobiol Dis* 98: 77–87
- Hohnholt MC, Andersen VH, Andersen J V, Christensen SK, Karaca M, Maechler P & Waagepetersen HS (2018) Glutamate dehydrogenase is essential to sustain neuronal oxidative energy metabolism during stimulation. *J Cereb blood flow Metab Off J Int Soc Cereb Blood Flow Metab* 38: 1754–1768
- Kim G-H, Lee Y-E, Lee G-H, Cho Y-H, Lee Y-N, Jang Y, Paik D & Park J-J (2015) Overexpression of malic enzyme in the larval stage extends *Drosophila* lifespan. *Biochem Biophys Res Commun* 456: 676–682
- Lennicke C & Cochemé HM (2020) Redox signalling and ageing: insights from *Drosophila*. *Biochem Soc Trans* 48: 367–377
- Luo W, Chen W-F, Yue Z, Chen D, Sowcik M, Sehgal A & Zheng X (2012) Old flies have a robust central oscillator but weaker behavioral rhythms that can be improved by genetic and environmental manipulations. *Aging Cell* 11: 428–438
- Rey G, Valekunja UK, Feeney KA, Wulund L, Milev NB, Stangherlin A, Ansel-Bollepalli L, Velagapudi V, O'Neill JS & Reddy AB (2016) The Pentose Phosphate Pathway Regulates the Circadian Clock. *Cell Metab* 24: 462–473
- Scialò F, Sriram A, Fernández-Ayala D, Gubina N, Löhmus M, Nelson G, Logan A, Cooper HM, Navas P,

- Enríquez JA, *et al* (2016) Mitochondrial ROS Produced via Reverse Electron Transport Extend Animal Lifespan. *Cell Metab* 23: 725–734
- Sunderhaus ER, Law AD & Kretschmar D (2019) ER responses play a key role in Swiss-Cheese/Neuropathy Target Esterase-associated neurodegeneration. *Neurobiol Dis* 130: 104520
- de Tredern E, Rabah Y, Pasquer L, Minatchy J, Plaçais P-Y & Preat T (2021) Glial glucose fuels the neuronal pentose phosphate pathway for long-term memory. *Cell Rep* 36: 109620
- Ugur B, Bao H, Stawarski M, Duraine LR, Zuo Z, Lin YQ, Neely GG, Macleod GT, Chapman ER & Bellen HJ (2017) The Krebs Cycle Enzyme Isocitrate Dehydrogenase 3A Couples Mitochondrial Metabolism to Synaptic Transmission. *Cell Rep* 21: 3794–3806
- Volkenhoff A, Weiler A, Letzel M, Stehling M, Klämbt C & Schirmeier S (2015) Glial Glycolysis Is Essential for Neuronal Survival in *Drosophila*. *Cell Metab* 22: 437–447
- Vrila-Mortimer A, Gomez R, Dowse H & Sanyal S (2012) A survey of the protective effects of some commercially available antioxidant supplements in genetically and chemically induced models of oxidative stress in *Drosophila melanogaster*. *Exp Gerontol* 47: 712–722
- Yang Y, Ye T, Yu J, Fan L, Ma C, Zhang B & Tan T-C (2025) Evaluation of the antioxidant and longevity-promoting effects of white tea extract in *Drosophila melanogaster*. *Front Nutr* 12: 1702854

Dear Professor Schirmeier,

Re: JP-RP-2026-288582R1 **"Neuronal loss of the Pentose Phosphate Pathway in the living nervous system is causally linked to [NADPH] reduction and elevated oxidative stress"** by Stephan Müller, Nina Surina, Andrés Köhler-Solís, Ioannis Nellas, Astrid Fleige, Sebastian Görtz, and Stefanie Schirmeier

We are pleased to tell you that your paper has been accepted for publication in The Journal of Physiology.

Yours sincerely,

Katalin Toth
Senior Editor
The Journal of Physiology

IMPORTANT POINTS TO NOTE FOLLOWING ACCEPTANCE OF YOUR PAPER:

- **IMPORTANT NOTICE ABOUT OPEN ACCESS:** To assist authors whose funding agencies mandate immediate public access to published research findings, The Journal of Physiology allows authors to pay an Open Access (OA) fee to have their papers made freely available immediately on publication.

- You can help your research get the attention it deserves! Check out Wiley's free Promotion Guide for best-practice recommendations for promoting your work at: www.wileyauthors.com/eoo/guide. You can learn more about Wiley Editing Services which offers professional video, design, and writing services to create shareable video abstracts, infographics, conference posters, lay summaries, and research news stories for your research at: www.wileyauthors.com/eoo/promotion.

- If you would like to receive our 'Research Roundup', a monthly newsletter highlighting the cutting-edge research published in The Physiological Society's family of journals (The Journal of Physiology, Experimental Physiology, Physiological Reports, The Journal of Nutritional Physiology and The Journal of Precision Medicine: Health and Disease), please click this link, fill in your name and email address and select 'Research Roundup': <https://www.physoc.org/journals-and-media/membernews>

EDITOR COMMENTS

Reviewing Editor:

Thank you for the final minor revisions that you have made in the manuscript. All concerns have been adequately addressed.

REFEREE COMMENTS

Referee #2:

The authors have now addressed my comments in a satisfactory manner and performed the appropriate changes in the current version of the manuscript. While some suggested experiments were not feasible, the authors provide clear justifications and discuss the imposed limitations. Despite this, the authors now added a novel behavioral analysis which in my view further strengthens the manuscript findings and concepts and extends its relevance beyond structural neurodegeneration.

I found it interesting that the authors report that while antioxidant treatment ameliorated the neurodegenerative phenotype caused by neuronal PPP knockdown, it failed to rescue the age-dependent behavioral alterations. In my view, instead of a failed experiment, these findings might indeed be informative - they may suggest that increased oxidative stress and impaired redox homeostasis are likely mediators of neuronal degeneration, whereas the behavioral changes may derive from other PPP-dependent metabolic functions. Thus, beyond its role in NADPH production and ROS detoxification, PPP activity may be required to sustain proper neuronal activity or circuit function during aging, potentially through effects on biosynthetic pathways or neuronal metabolism that are not compensated by antioxidant supplementation. These points could be considered for discussion in the Discussion section, if the data is robust and the editor feel they would add value to the manuscript.

Overall, I commend the authors for their efforts in addressing my comments in a thorough and transparent manner.